

# Evidence for stronger discrimination between conspecific and heterospecific mating partners in sexual *vs.* asexual female freshwater snails

Sydney Stork[1], Joseph Jalinsky[1] and Maurine Neiman[1,2]

[1] University of Iowa, Iowa City, United States
[2] Department of Gender, Women's, and Sexuality Studies, University of Iowa, Iowa City, USA

Corresponding author
Maurine Neiman,
maurine-neiman@uiowa.edu

## ABSTRACT

Once-useful traits that no longer contribute to fitness tend to decay over time. Here, we address whether the expression of mating-related traits that increase the fitness of sexually reproducing individuals but are likely less useful or even costly to asexual counterparts seems to exhibit decay in the latter. *Potamopyrgus antipodarum* is a New Zealand freshwater snail characterized by repeated transitions from sexual to asexual reproduction. The frequent coexistence of sexual and asexual lineages makes *P. antipodarum* an excellent model for the study of mating-related trait loss. Under the presumption (inherent in the Biological Species Concept) that failure to discriminate between conspecific and heterospecific mating partners represents a poor mate choice, we used a mating choice assay including sexual and asexual *P. antipodarum* females and conspecific (presumed better choice) *vs.* heterospecific (presumed worse choice) males to evaluate the loss of behavioral traits related to sexual reproduction. We found that sexual females engaged in mating behaviors with conspecific mating partners more frequently and for a greater duration than with heterospecific mating partners. By contrast, asexual females mated at similar frequency and duration as sexual females, but did not mate more often or for longer duration with conspecific *vs.* heterospecific males. While further confirmation will require inclusion of a more diverse array of sexual and asexual lineages, these results are consistent with a scenario where selection acting to maintain effective mate discrimination in asexual *P. antipodarum* is weak or ineffective relative to sexual females and, thus, where asexual reproduction is associated with the evolutionary decay of mating-related traits in this system.

## INTRODUCTION

Adaptations that were once important for survival but subsequently become useless or impose new fitness costs tend to decay over time in a process known as vestigialization (*Darwin, 1859*; *Wiedersheim, 1895*; *der Kooi & Schwander, 2014*). Many examples of this phenomenon can be seen throughout evolutionary history, including the loss of limbs in

snakes (*Lande, 1978*), eye structures in cave-dwelling fish (*Jeffery, 2005*), and wings in many insect species (*Roff, 1990*).

While the genetic mechanisms underlying vestigialization depend on evolutionary context, traits are often lost *via* the accumulation of selectively neutral mutations (*Fong, Kane & Culver, 1995*; *Lahti et al., 2009*). The decay of presently useless characteristics exemplifies evolutionary tradeoffs: the decay of one trait frees up energy and resources to be used elsewhere by the organism (*Fong, Kane & Culver, 1995*). For example, *Roff (1990)* suggests that the evolution of flightlessness from flighted ancestors allows for allocation of resources for reproduction. This framework implies that trait loss could translate into increased fitness for organisms in which the lost trait is no longer useful.

The decay of traits specific to mating and/or sexual reproduction (and above and beyond the loss of fitness *via* the accumulation of harmful mutations expected under asexual reproduction (*Muller, 1964*; *Hill & Robertson, 1966*)) is one potential consequence of transitions from sexual to asexual reproduction. These traits are typically required for reproductive success in sexual organisms. By definition, asexual organisms do not need external genetic contributions to reproduce, with the exception of sperm-dependent forms of parthenogenesis (*e.g.*, gynogenesis) (reviewed in *Neiman, Sharbel & Schwander, 2014*). Mating behaviors are themselves associated with numerous costs (*Williams, 1966*; *Magnhagen, 1991*). Accordingly, eventual vestigialization of mating-related traits in asexual lineages—including those involved in mate choice—is expected, particularly in a scenario where such behaviors and their maintenance are costly (*Carson, Chang & Lyttle, 1982*; *der Kooi & Schwander, 2014*; *Kampfraath et al., 2020*). Alternatively, these traits may be selectively neutral or maintained by selection in asexuals, but only in a context where the costs do not outweigh the benefits of trait maintenance (*Schwander et al., 2013*; *Kraaijeveld et al., 2016*). Several studies have supported these ideas, finding abnormal or decayed traits involved in mating and reproduction (spermathecae in asexual *Timema* stick insects (*Schwander et al., 2013*); sperm in asexual *Potamopyrgus antipodarum* males (*Jalinsky, Logsdon & Neiman, 2020*)) in asexual organisms. These results suggest that these traits are not being maintained by selection, at least to the extent of the same traits in sexual counterparts. Our focus is on behaviors: under the assumptions that mate preferences are associated with mate quality (*Zahavi, 1975*; *Kirkpatrick, 1982*; *Heywood, 1989*) (*vs.*, *e.g.*, sensory drive; *Endler, 1992*), that mate quality contributes to fitness of sexual but not asexual females, that the genes underlying mating behaviors are not pleiotropic with respect to other traits also important to asexual females, and that there is at most limited gene flow between sexual and asexual conspecifics, we ask whether mating-related behaviors associated with mate preference decay in asexual females relative to sexual counterparts.

*Potamopyrgus antipodarum* is a New Zealand freshwater snail characterized by the frequent coexistence of phenotypically and ecologically similar obligately sexual and obligately asexual lineages (*Lively, 1987*; *Wallace, 1992*). Transitions to obligately asexual reproduction from obligately sexual conspecific *P. antipodarum* appear to be one-way and irreversible (*Dybdahl & Lively, 1995*; *Paczesniak et al., 2013*), making this species an

excellent model to study the vestigialization of traits involved in sexual reproduction (*e.g.*, *Jalinsky, Logsdon & Neiman, 2020*).

Female asexual *P. antipodarum* mate readily with conspecific males (*e.g.*, *Neiman & Lively, 2005*; *Neiman et al., 2011*; *Soper, Hatcher & Neiman, 2015*), though there is no evidence that these females experience a benefit of copulation (*Neiman, 2006*). Previous studies have shown that mating frequency or duration in sexually reproducing male *P. antipodarum* does not appear to differ from that of the presumed asexual males produced occasionally by asexual females (*Soper, Hatcher & Neiman, 2015*). While gene flow mediated by these "asexual" males is formally possible (*Neiman et al., 2011*), separate lines of evidence for distinct genomic consequences of asexuality in *P. antipodarum* (*Sharbrough et al., 2018*; *McElroy et al., 2021*) suggests that such gene flow is rare at most. Sexual males also show poor ability to discriminate between favorable and unfavorable mates, mating with equal frequency with sexual and asexual females and with parasitically castrated *vs.* healthy females (*Neiman & Lively, 2005*). These previous results set the stage for a direct comparison between the strength of mate choice in sexual and asexual *P. antipodarum* females. In a mating paradigm like that demonstrated for *P. antipodarum* where males seem to lack strong mate choice behavior, differences in mating between sexual and asexual females, and especially in an experimental setting where the male lacks choice, can reasonably be attributed to mating differences between these two female groups. Therefore, we hypothesize that sexual female *P. antipodarum* are better able to choose a more favorable mate when presented with a choice than asexual counterparts.

We evaluated this hypothesis by comparing the mating behavior of sexual and asexual *P. antipodarum* females with sexual male *P. antipodarum* and sexual male *Potamopyrgus estuarinus*. *Potamopyrgus estuarinus* is a closely related and phenotypically similar species that nevertheless has minimal habitat overlap (and, to our knowledge, no evidence for sympatry) with *P. antipodarum* (*Haase, 2008*). Under the Biological Species Concept (*Mayr, 1942*), mating efforts with conspecific males are presumed to be favorable relative to mating with heterospecifics because of the possibility to generate offspring only with the former. Preference for *P. antipodarum vs. P. estuarinus* should be selectively neutral for asexual female *P. antipodarum*, for which eggs develop in the absence of males and fertilization. We chose to use *P. estuarinus* males as a heterospecific (poor) alternative choice in this experiment because the two species are phenotypically similar and because mating behavior (but not offspring production) has been shown to occur between male and female *P. antipodarum* and *P. estuarinus* (S. Stork, 2019, unpublished data), thereby providing *P. antipodarum* females with what reasonably appears to be a poor alternative choice. We used an experimental design where individual females were housed with a male from each species, meaning that only females had the opportunity to choose between mates.

## MATERIALS AND METHODS

### Snail selection and care

We haphazardly selected three groups of ten sexual ($N = 30$) and asexual ($N = 30$) female *P. antipodarum* descended from one sexual and one asexual female, respectively, collected from Lake Mapourika, New Zealand, in 2018. These six groups of 10 female snails will

hereafter be referred to as 'populations.' Because asexual female *P. antipodarum* are usually recently derived from sympatric sexual counterparts (*Dybdahl & Lively, 1995*; *Paczesniak et al., 2013*), trait values measured from sexual females provide an appropriate baseline for comparison to the values of the same trait in sympatric asexual females. We followed the lead of many other *Potamopyrgus antipodarum* studies (*e.g.*, *Krist et al., 2014*) that use data on the relationship between female *P. antipodarum* length and reproductive activity (*e.g.*, *Richards & Shinn, 2004*; *McKenzie, Hall & Guralnick, 2012*; *Larkin, Tucci & Neiman, 2016*) to establish three millimeters in length as a proxy for reproductive maturity for both sexual and asexual females. We also haphazardly selected three groups of 20 each sexually mature (possessing a visible penis, *Haase, 2008*) sexual *P. antipodarum* males ($N = 60$) and *Potamopyrgus estuarinus* males ($N = 60$). All *P. antipodarum* males were chosen from one of three sexual lineages independently established by three different sexual females sampled from New Zealand lakes Mapourika and Selfe in 2018. These sexual lineages from which we drew males were derived from different females than the sexual lineages from which we drew females. All *P. estuarinus* males were collected from the Ashley River estuary in New Zealand in February, 2020.

All snails used in the experiment had been housed in 10 L plastic tanks filled about 75% full with carbon-filtered tap water and fed *Spirulina* spp. algae, a common *Potamopyrgus* spp. lab food, three times weekly. These tanks were maintained in our constant-temperature snail room, which is held at 16 °C and on a 12:12 h light:dark cycle. Snails were given powdered chalk as a source of dietary calcium once weekly (feeding and maintenance following *Nelson & Neiman (2011)*). We isolated each population used for the experiment in a 0.95 L plastic cup containing ~700 mL of water for approximately 14 days prior to behavioral trials to control for previous exposure to males and possible recent mating. We painted female shells with light-colored nail polish to aid in differentiation between females and males when analyzing video footage. Feeding and maintenance for the populations in cups followed our standard snail room protocol.

Mating trials occurred in standard four-inch petri dishes. We covered the exterior of each dish in electrical tape to control for external visual stimuli. To begin each trial for the sexual females, we placed one sexual *P. antipodarum* female, one sexual *P. antipodarum* male, and one *P. estuarinus* male equidistant from one another around the edges of the inside of the petri dish (Fig. 1). After the three snails were placed, we filled the petri dish to about 75% total volume with carbon-filtered tap water. We used this same procedure for asexual *P. antipodarum* females, with the asexual female replacing the sexual female. No snail was used in more than one trial, and all 30 asexual females, all 30 sexual females, and all 60 males described above were used in the experiment.

## Observation of mating behavior

We conducted each mating trial at approximately 15:00 h, for three hours. As the maximum copulation duration for *P. antipodarum* is about 2 h (*Neiman & Lively, 2005*), we established this 3-h time frame to allow for multiple mating attempts to occur. We recorded trials using a USB webcam and time-lapse video software VideoVelocity (v3.7.2090, *CandyLabs, 2019*). Similar to the open-format approach taken in previous

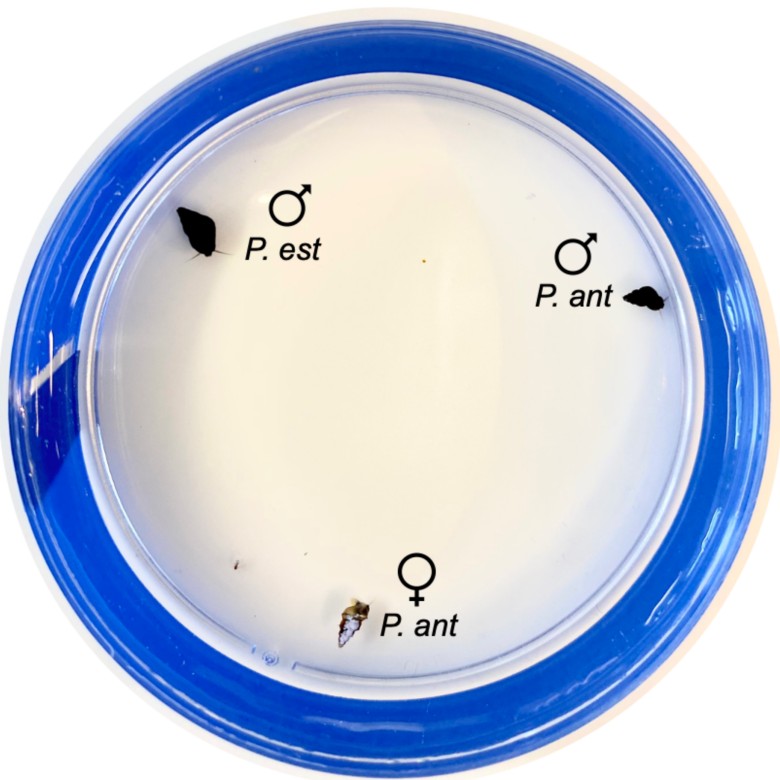

**Figure 1 Snail placement for mating trials.** One *P. estuarinus* male ("*P. est* " + male symbol), one *P. antipodarum* male ("*P. ant*" + male symbol), and one sexual or asexual *P. antipodarum* female ("*P. ant*" + female symbol) were placed approximately equidistant from one another around the edges of the petri dish. Each female was painted a light color with nail polish.

studies of mating behavior in *P. antipodarum* (*e.g.*, Neiman & Lively, 2005; Neiman, 2006), the three snails per trial were allowed to move and interact freely without physical barriers.

We loosely followed previous studies of mating in *P. antipodarum* (*e.g.*, Nelson & Neiman, 2011) to define attempted mating behavior as any physical interaction resulting in one snail mounting the shell of another. It is otherwise impossible to visually confirm sexual contact (joining of genitalia) in *Potamopyrgus* because this phenomenon is internal relative to the operculum and is thus obscured by the shells. Once this mounting behavior was initiated, we arbitrarily established that any such interaction lasting 15 s or longer would be counted as an instance of mate choice; interactions shorter than 15 s were potentially too short to be discernible with our time-lapse software and were excluded. Fifteen seconds is also more than enough time for snails to reorient and move on following an incidental—*vs.* mating—encounter. Both the frequency (number of mating interactions each female had with conspecific *vs.* heterospecific males) and duration (length of time (seconds) that each female was in physical contact with each male type) of mating interactions between females and conspecific *vs.* heterospecific males were recorded. Females that did not mate at all during the 3-hour trial were given values of "zero" for all aspects of mating frequency and mating duration data (see Supplemental Tables).

We also used this experimental framework to establish expected outcomes and determine how these outcomes would inform our hypotheses. First, a fair test of the trait decay hypothesis would require that sexual females mate more frequently and/or for longer with conspecific males than with *P. estuarinus* males. This outcome is critical in indicating that sexual females seem to possess the ability to discriminate between preferential and poor mating partners in our experimental setting. Absence of a difference in mating behavior across male type for sexual females would not support the trait decay hypothesis, suggesting instead, for example, that these females are unable to differentiate between mates or that observed contact is reflective of behavior outside of mating. Mating-related trait decay in asexual females consistent with the trait decay hypothesis would be reflected in relatively weak or absent mate choice for conspecific *vs.* heterospecific males relative to choices in sexual female *P. antipodarum*.

### Statistical analyses

We used the Shapiro-Wilk test to determine that both the mating frequency (sexual females: $W = 0.769$, $p = 8.18 \times 10^{-8}$; asexual females: $W = 0.810$, $p = 1.12 \times 10^{-5}$) and the mating duration (sexual females: $W = 0.672$, $p = 1.05 \times 10^{-9}$; asexual females: $W = 0.669$, $p = 3.17 \times 10^{-8}$) datasets violated normality assumptions of parametric analysis. Accordingly, we used nonparametric approaches to analyze the data. Analyses for mating duration were executed using the total mating duration for each individual (Tables S1 and S2). First, we used two-tailed Mann-Whitney $U$ tests to compare overall mating frequency and duration between individual sexual and asexual females. Second, we used Wilcoxon signed-rank analyses to determine whether mating durations and frequencies of sexual and asexual females, respectively, with conspecific males differed from mating durations and frequencies with heterospecific males. As another means of comparing mating behavior across sexual and asexual females, we calculated discrimination scores—defined as the difference between mating frequency and mating duration, respectively, with conspecific *vs.* heterospecific males—for each female (Tables S3 and S4). As an example (Table S3), for female snail S1, the mating duration discrimination score equals the total mating duration score with the conspecific (*P. antipodarum* male; "PAM") of 242.72 s minus the total mating duration score with the heterospecific male (*P. estuarinus* male; "PEM") of 74.64 s, for a mating duration discrimination score of 168.08 s. We then used Mann-Whitney $U$ tests to determine whether there was a significant difference in the mating frequency and mating duration discrimination scores, respectively, between sexual and asexual females. All analyses were executed with R 3.6.2 (*R Core Team, 2017*), *via* the ggplot2 (v3.3.3; *Villanueva, Chen & Wickham, 2016*), ggpubr (v0.4.0; *Kassambara, 2020*) and dplyr (v1.0.4; *Wickham et al., 2021*) packages.

## RESULTS

Technical challenges with the recording software and camera hardware ultimately rendered data from three sexual female and ten asexual female trials impossible to extract, leaving us with trials from 27 sexual females and from 20 asexual females. Comparison of the mating frequency and duration of these sexual and asexual females revealed no

differences between the two female types for either trait (Frequency: Mann-Whitney $U = 252$, $z = -0.38$, $p = 0.704$; Duration: Mann-Whitney $U = 187$, $z = 1.78$, $p = 0.075$). Wilcoxon signed-rank analyses demonstrated that sexual females mated significantly more often with conspecific males (median = 1.0 matings, IQR = 2.0) than with heterospecific males (median = 1.0 matings, IQR = 1.0) ($W = 483$, $p = 0.030$) (Fig. 2A). By contrast, asexual females mated with conspecific males (median = 1.0 matings, IQR = 2.0) at the same frequency as with heterospecific males (median = 1.0, IQR = 2.5) ($W = 172$, $p = 0.427$) (Fig. 2A).

Sexual females also mated for significantly longer with conspecific males (median = 242.72 s, IQR = 1,447.82) than with heterospecific males (median = 22.76 s, IQR = 102.25) ($W = 529$, $p = 0.004$) (Fig. 2B). Asexual females did not differ in the duration of mating with conspecific (median = 30.65 s, IQR = 377.43) and heterospecific males (median = 62.50 s, IQR = 163.84) ($W = 203$, $p = 0.945$) (Fig. 2B).

The discrimination score analysis revealed that sexual females with a choice between conspecific and heterospecific males tend to make mating behavior decisions with respect to mating frequency that favor the former (Mann-Whitney $U = 175.5$, $z = 2.02$, $p = 0.043$), with no evidence for such discrimination in the asexual females. By contrast, there was no significant difference in discrimination scores for mating duration between sexual and asexual females (Mann-Whitney $U = 197$, $z = 1.56$ $p = 0.059$).

## DISCUSSION

Our experiment revealed that sexual female *P. antipodarum* exhibited significantly higher mating frequency and longer mating duration with conspecific than heterospecific males. These results are as predicted if sexual female *P. antipodarum* are able to discriminate between mates of higher *vs.* lower quality (as framed under the Biological Species Concept), though we cannot formally exclude the possibility that other factors that differ between male *P. antipodarum* and *P. estuarinus* like the larger size of the latter (*Haase, 2008*) or the potential for different mating behaviors between males from the two species played a role. These results are also important in demonstrating that behaviors that appear to represent better mate choices occur in the context of our experiment. By contrast, although asexual *P. antipodarum* females mated at the same frequency and for the same duration as their sexual counterparts (strikingly similar to results presented for asexual *vs.* sexual males in *Soper, Hatcher & Neiman (2015)*), we were not able to detect evidence for mate discrimination of preference in the mating behaviors of asexual females in our experiment. With the caveat that we were limited to the study of a single *P. antipodarum* asexual lineage, our findings are consistent with a scenario where sexual female *P. antipodarum* exhibit more distinct mate discrimination ability than their asexual counterparts.

Evolutionary decay of mating-related traits in an asexual context is a plausible explanation for these results and is consistent with a previous study demonstrating evidence for decay of sperm traits in the male offspring occasionally produced by asexual female *P. antipodarum* (*Jalinsky, Logsdon & Neiman, 2020*). Another non-mutually exclusive explanation for our results is provided by the possibility that asexual female

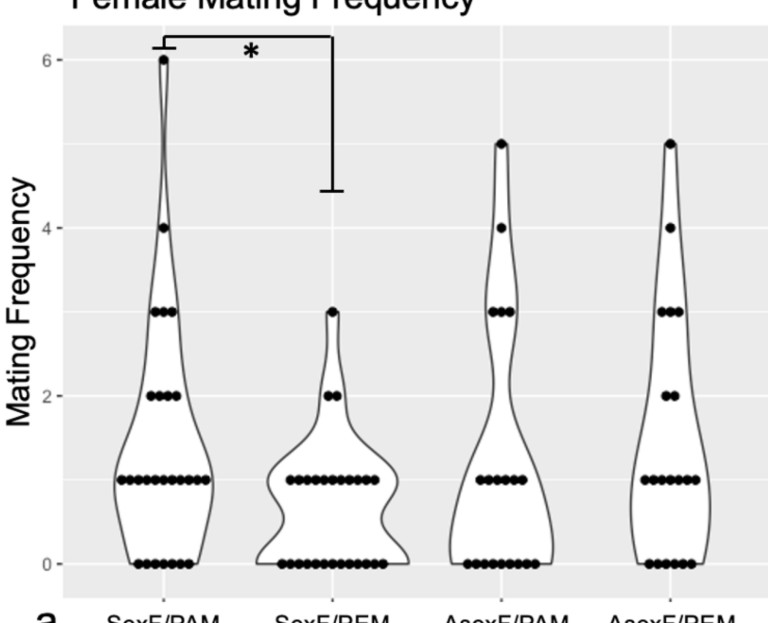

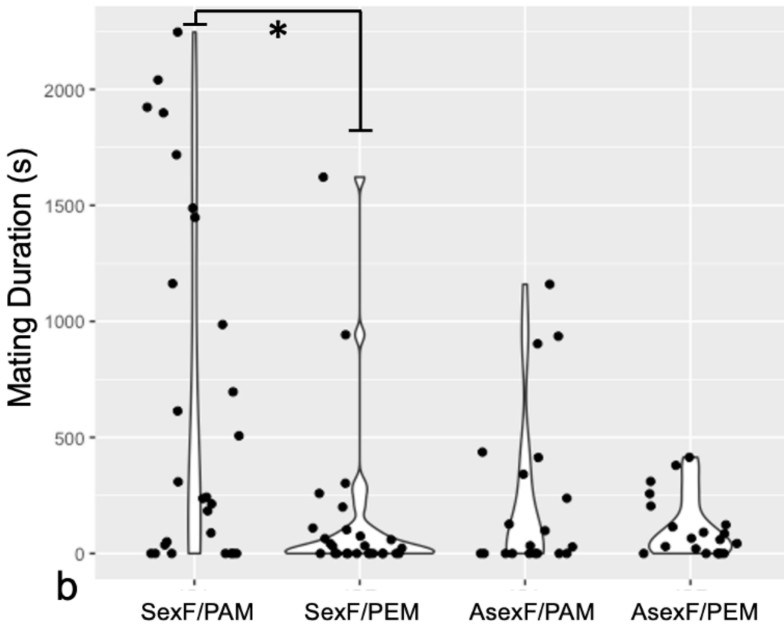

**Figure 2 Violin plots of (A) mating frequency and (B) total mating duration in sexual *vs*. asexual females with conspecific *vs*. heterospecific males.** Violin plots of mating frequency (A) and total mating duration (B) of sexual ("SexF") and asexual ("AsexF") females with *P. antipodarum* ("PAM") and *P. estuarinus* ("PEM") males. Individual data points are indicated with black dots. Statistically significant differences between pairwise comparisons ($p < 0.05$) are denoted with an asterisk (*).

*P. antipodarum* are experiencing more general degradation of phenotype associated with the accumulation of harmful mutations that is expected to accompany obligately asexual reproduction (*Muller, 1964*; *Hill & Robertson, 1966*). Indeed, a relatively high rate of accumulation of potentially harmful mutations has been observed in asexual *P. antipodarum* relative to sexual conspecifics (*e.g.*, *Sharbrough et al., 2018*; *McElroy et al., 2021*). Nevertheless, that asexual *P. antipodarum* females are similar to or even outperform sexual females when it comes to key determinants of fitness like fecundity (*Paczesniak et al., 2019*) and growth rate and age at reproductive maturity (*Larkin, Tucci & Neiman, 2016*) suggests that the expected mutation accumulation in the context of asexuality (and reported for asexual *P. antipodarum* in *Neiman et al., 2010*; *Sharbrough et al., 2018*) might not be translating into major phenotypic consequences. It is also important to consider that the lack of evidence for discrimination against mating with heterospecific males does not mean that asexual female snails might not demonstrate mating preferences in a different context (*e.g.*, amongst conspecific sexual males that differ in quality or between sexually *vs.* asexually derived males). Finally, it is important to acknowledge that our study only included female *P. antipodarum* from one lake and that we only compared one sexual lineage to one asexual lineage. Future studies using additional lineages will provide an critical additional line of evidence that our results are more broadly generalizable.

The exhibition of similar mating behavior to their sexual counterparts combined with the lack of detectable discriminatory behavior in asexual females could be explained by a situation where copulatory behavior—even in the absence of a connection between egg fertilization and embryo development—is still necessary and/or beneficial although *choice* of the mating partner is no longer important. For example, *Neiman (2004)* speculated that asexual females that have no direct use for sperm contributed by males might still benefit from mating if copulatory stimuli are required for maximization of reproduction, though *Neiman (2006)* showed that this scenario is unlikely for *P. antipodarum*. Alternatively, male ejaculate may provide nutritive benefits to asexual females. This possibility is supported by the fact that sperm storage structures are maintained in asexual *P. antipodarum* females (*Dillon, 2000*), though evidence that these structures might also function for the digestion of waste sperm in molluscs suggests the potential for a non-mating-specific or even unique function for this organ in asexuals (*Fretter & Graham, 1962*). Follow-up studies of whether and how sperm/ejaculate are used in asexual *P. antipodarum* and other asexual taxa will help clarify the prevalence and persistence of copulatory behavior in asexual females.

Another and perhaps simpler explanation is that mating behavior has not fully decayed in *P. antipodarum*, and that the ability to discriminate is weakened or lost before the behavior as a whole disappears. That asexual females nevertheless engage in mating behavior at all could be a function of the relatively recent derivation of most asexual female *P. antipodarum* from sexual counterparts (*Neiman, Jokela & Lively, 2005*). With this in mind, it would be valuable to repeat this experiment with females from older asexual lineages (*sensu Nelson & Neiman, 2011*), but departing from this prior study so that females *vs.* males have mate choice options and adequately controlling for size differences across lineages. Failure to discriminate between conspecific and heterospecific mates does

not necessarily mean that females will not discriminate between conspecific mates that vary in quality, such that an experiment comparing mate discrimination on a conspecific basis between sexual and asexual females will also provide an important step forward towards understanding how transitions to asexual reproduction influence subsequent evolution of mate choice behaviors.

## ACKNOWLEDGEMENTS

We thank the Iowa Center for Research by Undergraduates for contributing to mentorship and training for SS, Dr. Lori Adams for her input and guidance through the Honors Program for SS; John Logsdon, Mike Winterbourn, and Mary Morgan-Richards for help in snail collection; and Marissa Roseman for setting up, characterizing, and maintaining the snail lineages that we used. Josephine Bliss also contributed to snail care. We also acknowledge very constructive and helpful critiques from Prof. Ken Kraajiveld, Prof. Ingo Schlupp, Magdalena Mair, Alexandra Barnard, and several anonymous reviewers that greatly improved the paper.

### Funding

This work was supported by the Iowa Center for Research by Undergraduates, by Linda and Rick Maxson and *via* NSF grant DEB-1753851. The funders had no role in study design, data collection and analysis, decision to publish, or preparation of the manuscript.

### Grant Disclosures

The following grant information was disclosed by the authors:
Iowa Center for Research.
NSF: DEB-1753851.

### Competing Interests

The authors declare that they have no competing interests.

### Author Contributions

- Sydney Stork conceived and designed the experiments, performed the experiments, analyzed the data, prepared figures and/or tables, authored or reviewed drafts of the article, and approved the final draft.
- Joseph Jalinsky conceived and designed the experiments, authored or reviewed drafts of the article, and approved the final draft.
- Maurine Neiman conceived and designed the experiments, analyzed the data, authored or reviewed drafts of the article, and approved the final draft.

### Data Availability

All mating data used for data analyses are available in the Supplemental Files.
Supplemental snail videos are available upon request from the authors.

## Supplemental Information

Supplemental information for this article can be found online at http://dx.doi.org/10.7717/peerj.14470#supplemental-information.

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
