# Peer review of "Evidence for stronger discrimination between conspecific and heterospecific mating partners in sexual vs. asexual female freshwater snails"

_PeerJ, doi:10.7717/peerj.14470_

## Round 0.1 · original submission · Major Revisions

Thank you for your submission to PeerJ. Your manuscript has been thoughtfully reviewed by experts in the field who agree that the manuscript is clear, well written and interesting. They highlight several points that will improve your manuscript. Please carefully consider these points in your resubmission.

Reviewer 1 ·

Basic reporting

'no comment'

Experimental design

'no comment'

Validity of the findings

'no comment'

Additional comments

This is an interesting study, examining whether sexual and asexual individuals differ in their mate discrimination between conspecific and heterospecific partners, using the snail system. The writing is fluent, and the background literature is sufficiently provided to explain the context of the study objectives. The authors have also covered and discussed alternative explanations for the observed phenomena in the manuscript. The figures, tables, and supplementary material are good and necessary. I only have minor comments as follows:
1. The title, which is “Asexual freshwater snails make poor mate choice decisions”, is a little ambiguous because mating itself is a poor choice for asexual snails, and then it may not matter whether they mate with conspecific or heterospecific individuals.
It could be rephrased as, for example, “Sexual but not asexual freshwater snails differentiate between heterospecific and conspecific partners”.
2. Line 59: I would suggest replacing “for which” with “in which”
3. Line 63: Replace “(Muller 1964; Hill & Robertson 1966)” with “, Muller 1964; Hill & Robertson 1966”
4. Lines 65-67: Please add a reference for this sentence.
5. Lines 76-78: Another explanation for observing the traits could be a time lag in losing the traits, especially if they are not very costly?
6. Line 80: Please give reference for mate preferences driven by sensory drive
7. Lines 81-83: Please give references on which basis these assumptions are made “..that the genes underlying mating behaviors are not pleiotropic with respect to other traits also important to asexual females, and that there is at most limited gene flow between sexual and asexual conspecifics,..” so as to show that the assumptions do hold up in certain instances.
8. Line 91: Does “these males” refer to asexual males? Please clarify.
9. Line 107: Is the overlap in habitats of P. estuarinus with sexual or asexual lineages of P. antipodarum?
10. Lines 123-124, 130-131: Please clarify what “sets” refer to? It may not be clear to people not working with snails.
11. Lines 130: Is the three millimeters length as a proxy for reproductive maturity valid for both sexual and asexual lineages?
12. Line 134: What does “separately derived from the lineages from which we drew females” mean? Do you mean the lineages are from separate females or from the same females but at separate timepoints? Please clarify.
13. Lines 202-203: Please explain PAM and PEM, like it’s done in the Figure 2 legend.
14. Line 470: Please mention that females are of P. antipodarum.
15. Line 253-259: Please rephrase this long sentence to improve its clarity. I would suggest breaking it into sentences.

·

Basic reporting

Overall, this manuscript is clear and straightforward, although I see an issue with the underlying assumption, detailed below.

Suggestions to make the figures more clear:

Figure 1: rather than labeling snails A, B, and C, add male and female symbols and a species abbreviation for each male.

Figure 2: Combine A and B into a single plot, so the y-axis scale is the same for both sexual and asexual female results. This allows easier comparison between sexual and asexual females. Same for panels C and D: combine them into one. This will also allow you to indicate on the figure which comparisons were statistically significant.

The sentence on lines 227-229 is awkward and could be revised, but in general this was well-written.

Experimental design

The question of whether mate discrimination ability is reduced in asexual lineages compared to their sexual counterparts is an interesting one. However, this manuscript appears to conflate species recognition/hybridization avoidance with intraspecific mate choice. Mate choice is typically defined as being a within-species phenomenon. While it is true that mating with the wrong species is a worse choice than mating with a conspecific, it does not necessarily follow that females who avoid mating with heterospecifics are also capable of discriminating among male conspecifics based on quality. These two separate types of discrimination may rely on entirely different cues, and species recognition ability doesn't provide any information on a female's ability to make finer-scaled evaluations of conspecific male quality.

I think the experiment is well done and the question is interesting, but it needs to be reframed to reflect the distinction between intra- and interspecific discrimination.

I'm not sure if calculation and analysis of the mating frequency and duration scores is necessary when the original measures were already compared, especially since those scores weren't included in Figure 2. Either remove those or add more rationale (and maybe a figure) of what is added by this analysis.

Validity of the findings

See comments above. I think the findings are valid, but need some reframing or additional caveats.

Line 258: " [accumulated and potentially harmful] mutations might not be translating into major phenotypic consequences." This line needs clarification, since the potentially harmful mutations mentioned on line 254 aren't described. Do you mean major phenotypic consequences with respect to mate choice? Or more broadly?

Additional comments

As someone who doesn't know much about snails, I would appreciate a bit more information in the introduction about the asexual lineages. For example, what are the costs and benefits to mating for them? Why would an asexual female mate at all? This would help provide context for the experiment and results.

·

Basic reporting

In their manuscript „Asexual freshwater snails make poor mate choice decisions” Stork et al. present results from a study where they investigate the decay of the mate discrimination ability in an asexual lineage of the freshwater snail Potamopyrgus antipodarum.

The manuscript is very clearly written and the introduction, experimental procedures and results are presented in a concise and easy-to-understand way. The motivation to conduct the study is reasonably explained and sufficiently supported by the cited literature. The discussion addresses limitations of the study and discusses the results with respect to scientific literature.

Experimental design

The tested hypothesis is explicitly stated and the expected experimental outcomes are very well explained. This is in my experience not very commonly done and I appreciate this a lot.

The general experimental design fits to answering the proposed hypothesis, the procedures are clearly explained and the statistical analyses are described in sufficient detail to be replicated.

I have one major comment to the experimental design however, which should be taken into account when interpreting the results (see point 1 in the general comments below).

Validity of the findings

The raw data is provided in an accessible and easy-to-understand way. All abbreviations used in the raw data file are explained or deducible from the manuscript.
As a valuable additional supplement I would appreciate, if the authors could share their R code used for the statistical analyses as well.

In my opinion the interpretation of the study results in the context of the proposed hypothesis needs to be done more carefully, taking into account the limitations of the study design and statistical analysis (see general comments below).

Additional comments

I have two major points, which I think need to be considered and which might make a more careful interpretation of the study results necessary.

1. You state that all asexual females used in the experiments have been derived from the same asexually reproducing lineage. If I understand correctly, this means that all asexual females in the study are actually clones sharing the same genotype. If this is true, the difference between sexual and asexual females in the study could be simply a result of the relatively low mate choice ability of this one asexual lineage, and thus one female individual having started this lineage in the past. In consequence, an interpretation of the observed differences as being connected to asexual/sexual reproduction would be inappropriate. You mention this in the discussion, but I think this needs to be taken into account in all parts where the results are interpreted with respect to the proposed hypothesis, including the title (e.g. changing to “A lineage of asexual freshwater snails makes poor…”).

2. The second concern I have relates to the interpretation of non-significant p-values derived from the applied null hypothesis significance tests. Standard statistical tests as those used in the manuscript have a defined type I error rate (i.e., false positive rate) which is controlled by the alpha level threshold. The type II error rate (i.e., false negative rate) is however not controlled for. As a result, a non-significant result can be both, an indication for the true absence of an effect and the lack of statistical power. A non-significant result therefore remains always inconclusive and we cannot interpret this as absence of an effect.
For the manuscript this means that although the difference in mating frequency/ mating duration for asexual females was not significant, this does not mean that these females are unable to discriminate between con- and heterospecific males. Instead, the non-significance of the p-value could as likely be a result of sample size limitations, higher variance in these females’ mate choice ability, or a decreased, but still present, mate choice ability (smaller effect size).
Based on the provided data, I do believe that the asexual females in the study exhibited lower mate choice abilities than the sexual females, but I think this difference is gradual (making mistakes more often) rather than absolute (always making poor decisions, having lost or lacking mate choice ability).
Please check the manuscript again and adjust the interpretation of results where necessary (e.g., title, line 244, line 270, and maybe at other places I missed).

In addition I have some minor line by line comments that are not all essential, but which you might consider, if you agree:

Line 110: That mating with a heterospecific male does not lead to viable offspring is an important prerequisite for the present study. I suppose you are planning to publish these data (Storck, unpublished) somewhere else, but if possible, it would be great to have them cited here somehow. Either as a published article, a preprint, a data publication or (if not published anywhere else) as a raw data file in the supplement.

Lines 130-132: Is each set of PAM males derived from a single mother female? Or are these taken randomly from the established breeding lines? This is maybe explained in lines 133-134: “These lineages were separately derived from the lineages from which we drew females”, but this sentence is not entirely clear to me.

Line 137: Consider adding “spp.” to both Spirulina and Potamopyrgus.

Line 147 (or somewhere else appropriate): Please state again explicitly how many replicates you conducted, i.e. how many sexual and asexual females you tested in the bioassays? This information is implicitly given when assuming that all females taken from the lineages have been also used in experiments, but given that it might have happened that one or more of these individuals dropped out for some reason before being used in the bioassays, it would be easier for the reader to have the number of replicates stated clearly.

What have you done, if a female did not mate at all within the three hours of the experiment? This is clear when looking at the raw data, but would be good to be stated here explicitly.

Lines 188-192: I would suggest to remove lines 188-192. There is no need to justify the use of non-paramteric tests when considering the collected data structures:
Mating frequencies are counts of mating events and therefore should follow a Poisson distribution. If using a parametric test, you would have to fit a GLM/ Poisson regression and only check for the residuals to be normally distributed.
Concerning mating duration, the data are continuous, but cannot become negative. I would thus expect without conducting a Shapiro-Wilk test that mating durations do not follow a normal distribution, but instead are skewed with a long tail to the right and a steep tail to the left.

Line 192: Total mating duration is directly correlated to the number of matings that each female consented to. Please explain, why you chose to analyse the total mating duration and not the average mating duration of each female.
Please also explain, why you chose to include females that did not mate at all in the analysis of mating duration. By including non-mating females here, the effect of deciding to mate or not is absorbed into the time measurements.

Lines 193-194: Please explain in the previous paragraphs (e.g. after lines 183-185 and maybe also in the introduction), why you conducted this test. You explain why you test for decreased or absent mate choice in asexual females. But how does a generally higher mating frequency in one of the two groups (sexual vs. asexual females) relate to the hypothesis? Or why is this test necessary as a form of control or experimental validation?

Line 194: Remove “individual”.

Lines 194-197, and discrimination scores: In your analysis, you currently test first whether mate choice is present in sexual females and whether it is also present in asexual females (two separate tests for the two groups). In addition, you then calculate the discrimination scores and combine the measured mating values for con- and heterospecific males into one value that can then be compared directly between asexual and sexual females. The aim thus is to test for an interaction between female lineage and male species, asking the question: Is the difference in the response to con- and heterospecific males different among the two female groups? You might thus consider conducting a combined analysis instead of this step-wise approach by fitting GLMs with main effects and interaction term to the data.

Line 195: Wilcoxon rank-sum test or Wilcoxon signed-rank test? If I remember correctly, the rank sum test is for independent samples, whereas the signed-rank test is for paired samples (as should be used here).

Line 200: Include a reference to the supplementary tables S3 and S4 here. Reference to “Supplementary Table 3”is structured differently from the reference to“(Table S1, S2)” in line 193.

Line 207: Remove the second “the” before ggplot2.

Lines 227-229: This sentence is hard to read and does not exactly reflect what the test is testing for. I would suggest to rephrase this to: "...revealed that in terms of mating frequency, sexual females showed stronger preference for conspecific males than asexual females (test results)."

Lines 235: Consider replacing “vs.” with “than”.

Lines 239-241: This sentence seems redundant to me.

Line 244 – “no evidence for”: Please rephrase to interpret the results more carefully.
Considering the collected data, it is not possible to conclude that asexual females have no mate discrimination ability at all, or that there is no evidence that they have mate discrimination abilities. When looking at the mating duration data for example, asexual females seem to mate longer with conspecifics than with heterospecific males. Although the p-value for this comparison is non-significant, this does not allow to conclude that there is no difference.

Lines 259-260: This is a very important point. How likely is it in a natural environment that an asexually reproducing female is getting into contact with a sexually produced male? Can asexual females become sexual again?
As a non-expert in snail biology, I would appreciate a lot if you could add one or two sentences on the snails' life cycle and whether or how switches between sexual and asexual reproduction can occur in the introduction or discussion part.

Lines 265-267: This is a very important point!
And it raises the question whether the differences that were found between sexual and asexual females are indeed a result of sexual versus asexual reproduction, or whether all asexual females were just derived from one single female individual with comparably low mate discrimination abilities (see major comment above).

Line 270 – “lack of”: Please be more careful in interpreting the non-significant test outcomes. Non-significance does not allow for the interpretation of non-existence or lack of observable discriminatory behavior. In my opinion, based on the collected data, the discriminatory behaviour of the asexual females is very well observable, even if not significant.

Lines 275-276: Why is this scenario unlikely for P. antipodarum? Please explain.

Lines 290-291: It is not clear to me at this point, why these would be necessary improvements of the study. What would you learn, if you added male mate choice to the setup? Why do you think controlling for size differences would be important? Please explain and share your experience shortly.
Consider changing: “females and males” instead of “females vs. males”; “options” instead of “capabilities”.

Line 279: What is meant by waste here?

Lines 284-286: Great.

Figure 2:
Consider adjusting the lengths of the y-axes to be the same for a and b, and c and d, respectively. This is not necessary, but would give the reader the possibility to visually compare sexual and asexual females on a glance.

For SexF/PEM: Why are there 13 females that did not consent to matings (0 matings) in a, but about 20 females for which the total duration of mating was 0 in c?
For Asex/PAM: Here similarly, there are 9 females that had 0 matings in b, but 11 females that mated for 0 seconds.
Also for SexF/PAM, the numbers do not match.
This would mean, that there were females that did mate, but only for 0 seconds. Please explain or recheck the data and analysis.

From the figure it seems that mating duration was not measured continuously, but instead follows discrete steps. Please explain in the methods section, why this is the case.
In the raw data, the values are continuous. Please check whether something happened in the analysis steps. Maybe this happens when drawing the violin plots?

---

## Round 0.2 · Minor Revisions

Thank you for submitting this final version to PeerJ. It is now in a form that is ready for publication, barring one final comment from Reviewer 2 about the choice of points within the violin plots. Before this is accepted, please consider the reviewer's comments/suggestions

·

Basic reporting

This paper is clearly written and the authors made several changes suggested by reviewers to make the information even more clear.

Experimental design

The design of the experiment is sound, clearly described, and clearly displayed in Figure 1.

Validity of the findings

The authors revised graphs based on reviewer comments and reframed the experiment in a way that I believe better fits the experiment.

Additional comments

I appreciate the authors' thoughtful responses and revisions and I recommend that this paper be accepted for publication.

·

Basic reporting

no additional comment

Experimental design

no additional comment

Validity of the findings

no additional comment

Additional comments

I would like to thank the authors for considering my concerns and comments in their revised manuscript. The limitation of having tested only one asexual lineage is now stated clearly at several points in the manuscript, and non-significant results are now interpreted more cautiously. I appreciate the changes the authors made and also their explanations to points where the authors decided to stick to the original form.

I have only a last minor suggestion left which concerns the violin plot for mating durations:
I would like to thank the authors for their explanations concerning the data points from females that consented to mating but are displayed as having a total mating duration of zero in the violin plot in Figure 2b. I now understand that the violin plot here assigns data points to a specified number of bins, resulting in data points with small mating duration being displayed as zeroes. This clarifies where these data points come from and why there are discrete steps in mating durations in the plot. It is still not clear to me however, which advantage this binning of data points has. Binning (or grouping) of data points always comes with a loss of information. I thus would appreciate if the authors would consider plotting the measured values directly (e.g. with geom_jitter) and thus keeping the whole information of the raw data visible. This could be done very quickly, does not change the results and would be an easy way for giving the reader a better perception of the underlying data. Thank you.

---

## Round 0.3 · accepted · Accept

Thank you for considering those final comments. Excellent work, the manuscript is now ready for publication.

---

## Author Rebuttal · Round 0.3

COLLEGE OF
LIBERAL ARTS & SCIENCES
**Department of Biology**
143 Biology Building
Iowa City, IA 52242-1324
319.335.1050   Fax 319.335.1069
www.biology.uiowa.edu

October 2022

Dear Editors,

We appreciate the positive responses to our revised MS and sole suggestion regarding unbinning Figure 2b, which we have followed. There is no need for point-by-point responses beyond appreciation for the careful review and our confirmation that we have followed the suggestion of reviewer 3 and Editor Brannelly regarding Figure 2.

Sincerely,
Maurine

Maurine Neiman, Ph.D.
Professor
Department of Biology
Department of Gender, Women's and Sexuality Studies
Provost Faculty Fellow for Diversity, Equity, and Inclusion
University of Iowa
Senior Editor & Preprint Editor, Proceedings of the Royal Society of London B
maurine-neiman@uiowa.edu
http://bioweb.biology.uiowa.edu/neiman/
Twitter @mneiman
she/her

Editor comments (Laura Brannelly)
MINOR REVISIONS

Thank you for submitting this final version to PeerJ. It is now in a form that is ready for publication, barring one final comment from Reviewer 2 about the choice of points within the violin plots. Before this is accepted, please consider the reviewer's comments/suggestions

Reviewer 2 (Alexandra A Barnard)

[Figure]

**COLLEGE OF**
**LIBERAL ARTS & SCIENCES**

**Department of Biology**
143 Biology Building
Iowa City, IA  52242-1324
319.335.1050    Fax 319.335.1069
www.biology.uiowa.edu

Basic reporting

This paper is clearly written and the authors made several changes suggested by reviewers to make the information even more clear.

Experimental design

The design of the experiment is sound, clearly described, and clearly displayed in Figure 1.

Validity of the findings

The authors revised graphs based on reviewer comments and reframed the experiment in a way that I believe better fits the experiment.

Additional comments

I appreciate the authors' thoughtful responses and revisions and I recommend that this paper be accepted for publication.
* * *
Reviewer 3 (Magdalena M Mair)

Basic reporting

no additional comment

Experimental design

no additional comment

Validity of the findings

no additional comment

[Figure]

**COLLEGE OF**
**LIBERAL ARTS & SCIENCES**

**Department of Biology**
143 Biology Building
Iowa City, IA  52242-1324
319.335.1050    Fax 319.335.1069
www.biology.uiowa.edu

Additional comments

I would like to thank the authors for considering my concerns and comments in their revised manuscript. The limitation of having tested only one asexual lineage is now stated clearly at several points in the manuscript, and non-significant results are now interpreted more cautiously. I appreciate the changes the authors made and also their explanations to points where the authors decided to stick to the original form.

I have only a last minor suggestion left which concerns the violin plot for mating durations: I would like to thank the authors for their explanations concerning the data points from females that consented to mating but are displayed as having a total mating duration of zero in the violin plot in Figure 2b. I now understand that the violin plot here assigns data points to a specified number of bins, resulting in data points with small mating duration being displayed as zeroes. This clarifies where these data points come from and why there are discrete steps in mating durations in the plot. It is still not clear to me however, which advantage this binning of data points has. Binning (or grouping) of data points always comes with a loss of information. I thus would appreciate if the authors would consider plotting the measured values directly (e.g. with geom_jitter) and thus keeping the whole information of the raw data visible. This could be done very quickly, does not change the results and would be an easy way for giving the reader a better perception of the underlying data. Thank you.